# Antibacterial efficacy of cold atmospheric plasma against *Enterococcus faecalis* planktonic cultures and biofilms *in vitro*

Felix Theinkom[1,2], Larissa Singer[1,2], Fabian Cieplik[2], Sylvia Cantzler[3], Hannes Weilemann[3], Maximilian Cantzler[3], Karl-Anton Hiller[2], Tim Maisch[1☯‡]*, Julia L. Zimmermann[4☯‡]

1 Department of Dermatology, University Hospital Regensburg, Regensburg, Germany, 2 Department of Conservative Dentistry and Periodontology, University Hospital Regensburg, Regensburg, Germany, 3 terraplasma GmbH, Garching, Germany, 4 terraplasma medical GmbH, Garching, Germany

☯ These authors contributed equally to this work.
‡ These authors share senior authorship
* tim.maisch@ukr.de

**Data Availability Statement:** All relevant data are within the paper and its Supporting Information files.

## Abstract

Nosocomial infections have become a serious threat in our times and are getting more difficult to handle due to increasing development of resistances in bacteria. In this light, cold atmospheric plasma (CAP), which is known to effectively inactivate microorganisms, may be a promising alternative for application in the fields of dentistry and dermatology. CAPs are partly ionised gases, which operate at low temperature and are composed of electrons, ions, excited atoms and molecules, reactive oxygen and nitrogen species. In this study, the effect of CAP generated from ambient air was investigated against *Enterococcus faecalis*, grown on agar plates or as biofilms cultured for up to 72 h. CAP reduced the colony forming units (CFU) on agar plates by > 7 $\log_{10}$ steps. Treatment of 24 h old biofilms of *E. faecalis* resulted in CFU-reductions by $\geq 3$ $\log_{10}$ steps after CAP treatment for 5 min and by $\geq 5$ $\log_{10}$ steps after CAP treatment for 10 min. In biofilm experiments, chlorhexidine (CHX) and UVC radiation served as positive controls and were only slightly more effective than CAP. There was no damage of cytoplasmic membranes upon CAP treatment as shown by spectrometric measurements for release of nucleic acids. Thus, membrane damage seems not to be the primary mechanism of action for CAP towards *E. faecalis*. Overall, CAP showed pronounced antimicrobial efficacy against *E. faecalis* on agar plates as well as in biofilms similar to positive controls CHX or UVC.

## Introduction

Within the last 10 years, treatment of infectious diseases such as wound infections of the skin and the mucous membranes has become increasingly more complicated and ineffective due to the emergence of drug-resistant bacteria. In the 2016 Review on Antimicrobial Resistance, an alarming scenario was built up presaging that by 2050 10 million people could die per year

**Funding:** The authors Sylvia Cantzler, Hannes Weilemann, Maximilian Cantzler are employees from terraplasma GmbH (Garching, Germany), and Julia L. Zimmermann is anemployee from terraplasma medical GmbH (Garching, Germany), which have developed the plasma device used in this study. This study was funded by the grant "BayMed: 41-6618c/272/1-MED-1507-0004" (Bayern Innovativ GmbH, Germany). Furthermore, Fabian Cieplik thanks for funding by the University Medical Center Regensburg (ReForM B program) the Deutsche Forschungsgemeinschaft (DFG, German Research Foundation; grant CI 263/1-3). Denise Muehler is gratefully acknowledged for her valuable help with the DNA release experiments. The funders had no role in study design, data collection and analysis, decision to publish, or preparation of the manuscript.

**Competing interests:** The study employed a plasma source which had been developed and patented by terraplasma GmbH. The prime interest of terraplasma was to determine the safety of the plasma source in an independent study, carried out without undue influence by the company so that it provides a valuable documentation. The scientific value of this work lies in the possibility for advancing plasma treatment and providing the quantitative background for this future application, as well as paving the way for a clinical study. In this sense the interest is complementary – there is no competing interest. This does not alter the authors' adherence to PLOS ONE policies on sharing data and materials.

caused by multidrug-resistant bacteria [1]. Eradication of multi-resistant superbugs like the so-called "ESKAPE"-pathogens is one of the major clinical challenges of the twenty-first century [2]. One of the pathogens being often causative for nosocomial infections like catheter-associated urinary tract infections is the Gram-positive coccus *Enterococcus faecalis* [3]. *E. faecalis* combines many virulence factors like intrinsic resistances to several classes of antibiotics [4]. Furthermore, it is known for its ability to gain resistance in a relatively short amount of time [5]. In addition, *E. faecalis* is able to form biofilms on both, inanimate and living surfaces which raises tolerance and decreases susceptibility against environmental influences or antimicrobial measures [6, 7].

In general, bacterial biofilms are found on natural surfaces as well as on artificial ones like inside of catheters or on other implants [8]. Bacteria in such a sessile biofilm-mode are known to be much more tolerant towards antimicrobial approaches than their planktonic counterparts [9]. A primary mechanism for this enhanced tolerance is the extracellular polymeric substance (EPS) which bacteria are embedded in and which can act as a diffusion barrier for antimicrobial agents [10].

Therefore, novel treatment modalities that are capable of killing pathogens in biofilms with less or even no risk of inducing resistances are desperately needed [11]. Accordingly, already in 2011 a review published by Karen Bush *et al.* in *Nature Reviews Microbiology* had pointed out that „investigation of novel non-antibiotic approaches for the prevention of and protection against infectious diseases should be encouraged and that such approaches must be high-priority research" [12]. Therefore, new approaches must be developed that are capable of eliminating bacteria successfully, independently of their resistance patterns [13]. In this light, a novel therapeutic option may be cold atmospheric plasma (CAP), which is a highly innovative technology due to the point that no resistances have been induced in bacteria against CAP so far [14]. CAPs are partly ionized gases (with a typical ionization fraction of one ion or electron per a billion neutral atoms or molecules) producing a reactive mix by interacting with the surrounding air and being composed of electrons, ions, neutrons, excited atoms and molecules, reactive oxygen and nitrogen species, and UV light [15]. Depending on the respective plasma source technology, the carrier gas (here: ambient air), the plasma operating parameters and the set-up modalities (like transportation mode, volume etc.), the composition as well as the concentrations of the produced plasma species vary. This means, that CAPs can be "designed" to a certain extent and that different compositions of reactive species can be produced by changing input parameters such as carrier gas, voltage, frequency etc. [16].

CAPs are known to effectively inactivate bacteria and fungi (independent of their resistance profiles towards conventional antimicrobials as well as on different surfaces and in biofilms), spores and viruses [17, 18]. Furthermore, it could be shown that CAP doses can be found where pathogens can be inactivated without mammalian tissues being harmed or without changing the characteristics of the respective materials that bacteria are attached to [19, 20]. Due to these results and, moreover, as CAP operate at low temperatures (5–10˚C above room temperature only), various applications in hygiene (*e.g.* sterilization of medical equipment, hand disinfection *etc.*) and in clinical practice (*e.g.* treatment of infected wounds) are currently under investigation in the plasma medicine community [21, 22].

Consequently, plasma medicine–an evolving research field investigating CAP and their potential application in hygiene and medicine–has attracted a lot of interest in the past few years. However, the actual mechanisms of the antimicrobial action of CAP are still mostly unknown [23]. Furthermore, the antimicrobial efficacy of CAP against microbial biofilms is still not sufficiently explored and only very few studies have been performed up to this point [24–28]. Recently, our group showed that DNA damage seems not to be the primary mechanism of bacterial cell death upon CAP treatment because *Deinococcus radiodurans*, which is

known to have a very efficient DNA repair mechanism, can easily be killed by CAP [29], whereas this bacterium survives ionizing radiation up to 5000 Gy without being harmed [30]. Taking into consideration that the external structures like cell walls and cytoplasmic membranes most likely are destroyed before nucleic acids can be targeted, it seems reasonable that these outer cellular structures may be first-line targets of CAP [31].

Therefore, the aims of this study were (1) to investigate the antimicrobial efficacy of CAP with a novel surface micro-discharge (SMD) technology in a thin-film design towards *E. faecalis* in planktonic cultures as well as in biofilms and (2) to evaluate whether CAP treatment leads to cytoplasmic membrane damage as measured by leakage of cytoplasmic components upon treatment.

## Material and methods

### Thin-film surface micro-discharge plasma source

In this study, a CAP plasma source prototype was used under ambient conditions (natural fluctuation room with ≈21˚C). This prototype consists of one surface micro-discharge (SMD) plasma source with thin-film design composed of an insulator plate made out of $Al_2O_3$ (thickness 0.25 mm), a high voltage (= HV) electrode coated with Cu and Sn (thickness of 3.0 μm and 1.0 μm, respectively) electrode and a structured electrode, meaning an electrically grounded stainless-steel mesh electrode (basic material 1.4310, thickness 0.5 mm, square-shape structured). The novel thin-film design is the result of further development of the SMD plasma source described by Shimizu et al. [32]. The new design provides a technique that can be manufactured in series with minimized height. Therefore, the plasma source only needs a minimum of power that allows for battery-driven operation. The plasma is produced homogenously on the structured electrode by applying high sinusoidal AC voltage of 3.5 $kV_{PP}$ at a frequency of 4.0 kHz between the HV electrode and the structured electrode. In contrast to DBD-technology (dielectric-barrier-discharge), there is no current through the treatment object. The power consumption of this plasma source is about 0.5 to 1 W (depending on the surrounding humidity) and contains mainly reactive oxygen species.

The plasma source (composed of HV electrode, insulator plate and structured electrode) itself is embedded in a plastic case to ensure safe and easy handling during the experimental procedures (Fig 1). The size of the plasma source including the outer rim is 39.5 mm x 39.5 mm. The size of the plasma production area on the structured electrode is 29.5 mm x 29.5 mm. A spacer (inner height 10 mm, inner diameter 40 mm, outer diameter 50 mm, treatment area 12.56 $cm^2$, material Polytetrafluorethylene) which is attached to the plasma source and surrounds the structured electrode, allows the formation of a (semi-) closed volume of 126 mL (between plasma source and biological sample) so that the produced plasma species are confined inside.

The plasma source is placed above the sample. The transportation of the generated plasma species to the sample is carried out by diffusion and thermal convection. Depending on the experiments, treatment periods ranged between 1 min and 10 min in this study.

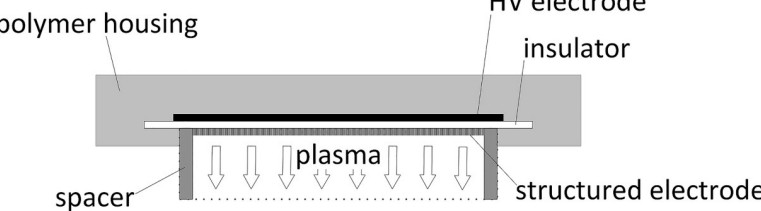

**Fig 1. Schematic view of the plasma source prototype with surface micro-discharge-technology (SMD) in thin-film design.** Ambient air is used and an area of interest of 12.56 $cm^2$ can be treated.

## Bacterial culture

*Enterococcus faecalis* (DSM 2570, ATCC 29212) was obtained from the DSMZ (Deutsche Sammlung von Mikroorganismen und Zellkulturen GmbH, Braunschweig, Germany) to be used for these experiments. *E. faecalis* was maintained on Müller-Hinton agar plates. For inoculation, one single colony was picked, suspended in 5 ml Brain Heart Infusion broth (BHI; Sigma-Aldrich, St. Louis, MO, USA) and incubated over-night at 37˚C on an orbital shaker (180 rpm). Afterwards, cultures were harvested by centrifugation (10 min, 3000 rpm; Table centrifuge Universal 320R, Hettich Germany), resuspended in phosphate-buffered saline (PBS; Dulbecco's Phosphate Buffered Saline, Sigma-Aldrich, St. Louis, MO, USA) and the optical density (OD) was measured at 600 nm by means of a spectrophotometer (SPECORD 50 Plus, Analytik Jena, Jena, Germany). Bacterial suspensions were adjusted to an OD of 0.6 for planktonic experiments or to an OD of 0.13 for biofilm formation, respectively.

## CAP treatment towards planktonic bacteria on agar plates

*E. faecalis* suspensions were used at an OD of 0.6. 100 μL of bacterial suspension were spread on Müller-Hinton agar plates and then exposed to CAP treatment (for 1, 3, 5 or 10 min), while the spacer mentioned above ensured a distance of 10 mm between plasma source and samples. After subsequent incubation of the agar plates at 37˚C for 24 h, colony forming units (CFUs) were evaluated. The area of interest was calculated based on the inner diameter of the spacer (40 mm) related to the whole area of the agar plates. For computation of $\log_{10}$-reduction rates, CFUs of serial dilutions of the original bacterial suspensions at OD = 0.6 were evaluated.

## Biofilm formation

*E. faecalis* biofilms were formed as it has been described earlier in detail [33, 34]. Briefly, over-night cultures of *E. faecalis* (OD = 0.13) were harvested by centrifugation and resuspended in the complete saliva (CS) broth originally described by Pratten *et al.* [35], which was modified by adding 0.1% (w/v) sucrose [33, 34]. 2 mL of this bacterial suspension in CS broth were mixed with 1 mL fetal calf serum (FCS; Pan-Biotech GmbH, Aidenbach, Germany) and added to sterile polystyrene petri dishes (35 mm x 10 mm; Primaria™ Easy Grip Cell Culture Dish, Corning, NY, USA). Biofilms were incubated at 37˚C under aerobic conditions without shaking for 24, 48 or 72 h, respectively. For biofilms cultured for more than 24 h, growth medium was changed every 24 h.

After the respective culture period, growth medium was carefully discarded with a pipette. Afterwards, the biofilms were treated with CAP for 1, 3, 5 or 10 min, respectively. For biofilm inactivation a spacer adjusted to a height of 20 mm was placed over the polystyrene petri dish and the CAP plasma source was placed on top of it. The biofilm was removed by scratching the biofilm from the bottom of the plastic well as well as by pipetting up and down repeatedly. Removal of all attached cells was checked by using light microscopic observation of the bottom of each petri dish. For separation of aggregated bacteria, the tubes were vortexed for 10 s and placed in an ultrasonic (35 kHz) water bath chamber (USR 30 H, Merck Eurolab GmbH, Grafing, Germany) for 10 min. Subsequently, serial tenfold dilutions were prepared (up to $10^{-16}$) and aliquots (3x 20 μL) were plated on Müller-Hinton agar plates according to the method described by Miles *et al.* [36]. CFU were evaluated after aerobic incubation for 24 h at 37˚C.

UVC radiation and treatment with chlorhexidine digluconate (CHX; Sigma-Aldrich, St. Louis, MO, USA) were used as positive controls. UVC radiation was performed with a DNA crosslinker operating at 254 nm (Biometra GmbH, Göttingen, Germany; radiation doses: 0.005 J/cm$^2$, 0.195 J/cm$^2$ or 0.26 J/cm$^2$). CHX was used at concentrations of 0.2% or 2% and incubation was carried out for 5 min.

## Spectroscopic measurements for release of nucleic acids upon CAP treatment

To evaluate whether CAP treatment leads to damage of cytoplasmic membranes, release of nucleic acids from the cytoplasm was measured spectroscopically at 260 nm as described earlier [37, 38]. Biofilms were cultured as described above and treated with CAP for 5 or 10 min, respectively. As a positive control, the biofilms were scraped, resuspended in PBS and incubated with 100 μL lysozyme (40,000 units/mg; Sigma-Aldrich, St. Louis, MO, USA) for 30 min at 37˚C. Then, 100 μL Proteinase K (7.0–14.0 units/mg; Sigma-Aldrich, St. Louis, MO, USA) and 200 μL 1% sodium dodecyl sulfate (Carl Roth GmbH + Co. KG, Karlsruhe, Germany) were added and incubated for another 30 min at 37˚C. The biofilms were brought to suspension in 1 mL PBS and transferred to 1.5 mL Eppendorf tubes. After sonication in an ultrasonic water bath chamber (Sonorex Super RK 102 H; 35kHz) for 10 min, the tubes were centrifuged (13.000 rpm; 5 min) and the supernatants were collected and evaluated for nucleic acid release by assessing the OD at 260 nm with a NanoDrop$^{TM}$ 2000 spectrophotometer (PEQLAB, Erlangen, Germany).

## Data analysis

CFU results are shown as medians, 1$^{st}$ and 3$^{rd}$ quartiles and were calculated using SPSS for Windows, v. 25 (SPSS Inc., Chicago, IL, USA) from the values of at least six independent experiments. In the figures, horizontal solid and dashed lines show 3 and 5 $\log_{10}$ steps reduction of CFU compared to the untreated control (7.3 x 10$^{10}$ CFU in median). Medians on or below these lines mean a bacterial reduction by 3 $\log_{10}$ ($\geq$99.9%) or by 5 $\log_{10}$ ($\geq$99.999%) or higher, respectively. According to the guidelines of infection control this means a biologically relevant antimicrobial activity or a disinfectant effect, respectively [39]. Results from spectroscopic measurements are depicted as medians, minima and maxima, and were calculated by SPSS from the values of three independent experiments at least.

# Results

## Antimicrobial assay towards planktonic *E. faecalis* on agar plates

Firstly, we investigated the antimicrobial efficacy of CAP towards planktonic *E. faecalis* on agar plates (Fig 2). The untreated controls (no CAP) exhibited 7.3 x 10$^{10}$ CFU in median. CAP treatment for 1 min led to a CFU reduction of $\geq$ 7 $\log_{10}$ steps. Longer CAP treatment periods of 3, 5 or 10 min did further increase the killing efficacy (reductions of $\geq$ 8 $\log_{10}$ steps (3 or 5 min) or $\geq$ 9 $\log_{10}$ steps (10 min), respectively).

## Antimicrobial assay towards *E. faecalis* biofilms cultured for 24 h

Secondly the antimicrobial efficacy of this new CAP device was evaluated towards *E. faecalis* growing as 24 h old biofilms (Fig 3). Untreated biofilms contained 1.9 x 10$^{14}$ CFU in median. CAP treatment for 5 min showed a CFU-reduction by $\geq$ 3 $\log_{10}$ steps, while CAP treatment for 10 min resulted in a reduction by $\geq$ 5 $\log_{10}$ steps. Reductions for 1 and 3 min of CAP treatment were < 3 $\log_{10}$ steps.

In addition, CHX and UVC radiation were evaluated as positive controls compared to the CAP treatment efficacy towards *E. faecalis* (Fig 4). Again, *E. faecalis* biofilms were cultured for 24 h. CHX 0.2% led to a CFU reduction of $\geq$3 $\log_{10}$ steps after 5 min while CHX 2% resulted in reduction by $\geq$ 9 $\log_{10}$ steps below the detection limit. A UVC radiation dose of 0.005 J/cm$^2$ reduced the CFU by $\geq$ 5 $\log_{10}$ steps. Higher doses of 0.195 J/cm$^2$ or 0.26 J/cm$^2$ reduced CFU by $\geq$ 9 $\log_{10}$ steps below the detection limit. Overall UVC radiation as a contact-free

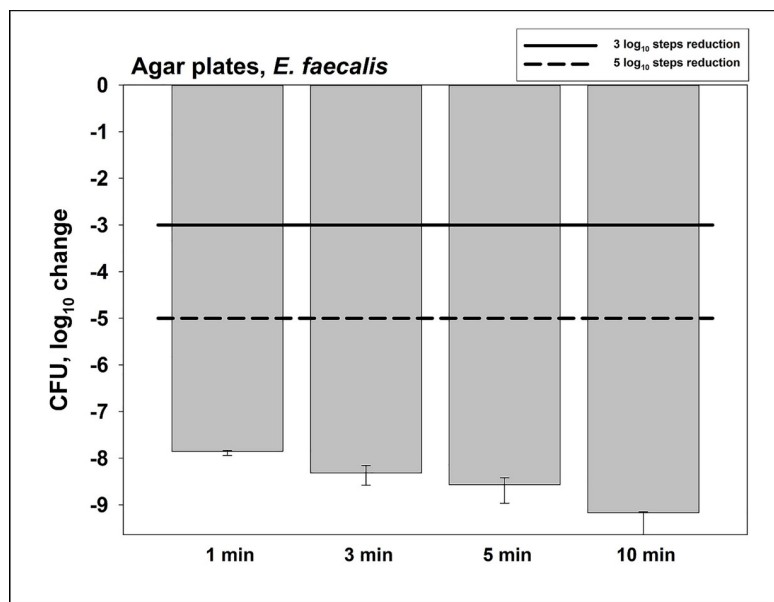

**Fig 2. Antimicrobial assay towards planktonic *E. faecalis* on agar plates.** CAP was performed for different treatment periods. Surviving colonies were counted 24 h later. All results are depicted as medians, $1^{st}$ and $3^{rd}$ quartiles from six independent experiments in duplicates on a $\log_{10}$-scaled ordinate. Bars show the reductions of CFU in a $\log_{10}$ scale in comparison to untreated controls. Solid and dashed lines exhibit 3 $\log_{10}$ steps (99.9%) or 5 $\log_{10}$ steps reduction (99.999%) of CFU, respectively. ($n = 6$; untreated controls exhibited $7.3 \times 10^{10}$ CFU in median).

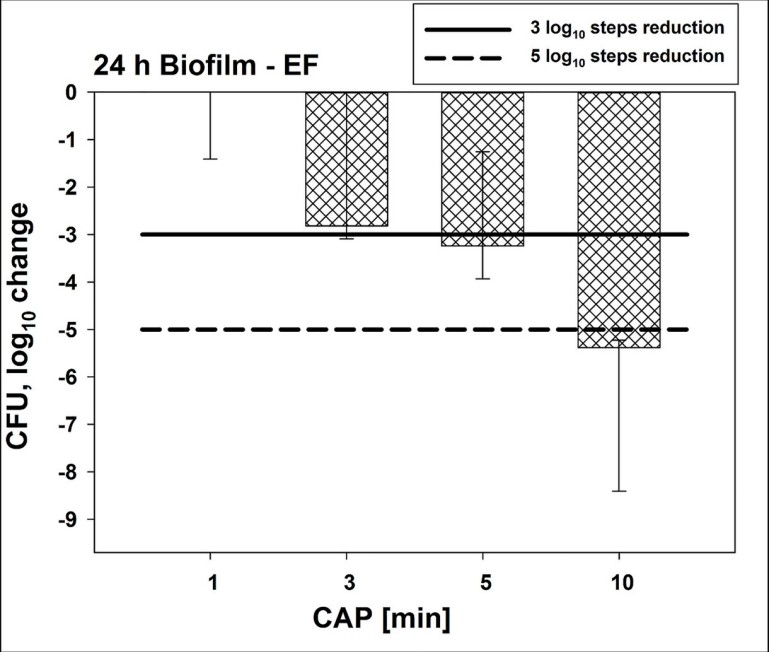

**Fig 3. Antimicrobial efficacy of CAP towards *E. faecalis* biofilms cultured for 24 h.** CAP was performed for different treatment periods towards biofilms cultured for 24 h. All results are depicted as medians, $1^{st}$ and $3^{rd}$ quartiles from six independent experiments in duplicates on a $\log_{10}$-scaled ordinate. Bars show the reductions of CFU in a $\log_{10}$ scale in comparison to untreated controls. Solid and dashed lines exhibit 3 $\log_{10}$ steps (99.9%) or 5 $\log_{10}$ reduction (99.999%) of CFU, respectively. ($n = 6$; untreated controls exhibited $7.3 \times 10^{10}$ CFU in median).

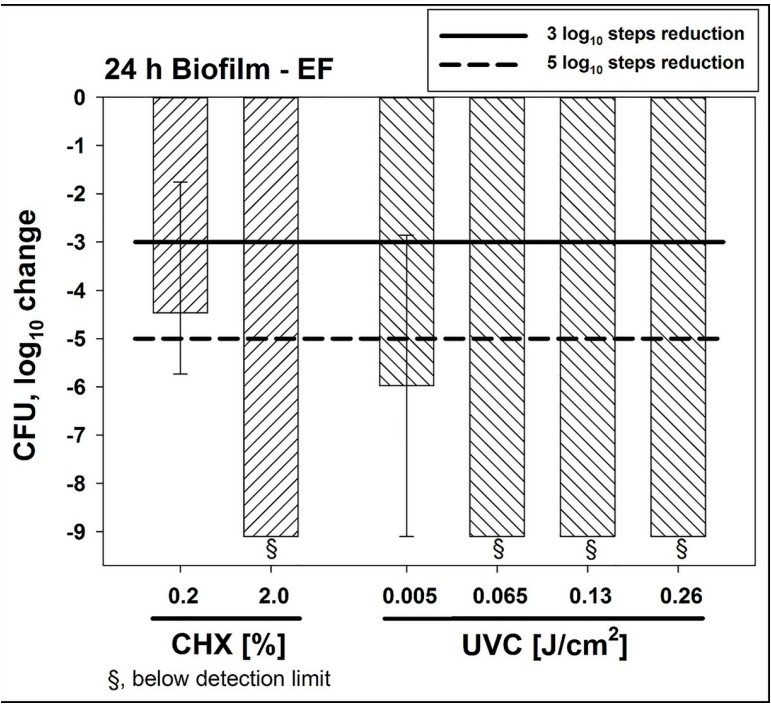

**Fig 4. Antimicrobial efficacy of positive controls CHX and UVC radiation towards *E. faecalis* biofilms cultured for 24 h.** Biofilms were incubated with CHX (0.2% or 2%) for 5 min. UVC radiation was performed applying distinct radiation doses (0.005 J/cm², 0.065 J/cm², 0.13 J/cm² or 0.26 J/cm²). All results are depicted as medians, 1st and 3rd quartiles from six independent experiments in duplicates on a log₁₀-scaled ordinate. Bars show the reductions of CFU in a log₁₀ scale in comparison to untreated controls. Solid and dashed lines exhibit 3 log₁₀ steps (99.9%) or 5 log₁₀ reduction (99.999%) of CFU, respectively. ($n = 6$; untreated controls exhibited $7.3 \times 10^{10}$ CFU in median).

application led to higher reduction of CFU in a dose-dependent manner compared to CAP, whereas the efficacy of CHX 0.2% treatment was similar to the CAP treatment.

## Antimicrobial assay towards *E. faecalis* biofilms cultured for 48 h or 72 h

The previous results emphasize to investigate the antimicrobial efficacy of CAP towards biofilms cultured for longer times like 48 h or 72 h (mature biofilms). Table 1 shows the results of CAP treatment, CHX or UVC radiation towards biofilms cultured for 48 h or 72 h. The CFU of *E. faecalis* were reduced by $\geq 5$ log₁₀ steps by a CAP treatment of 10 min for both 48 h and 72 h old biofilms. CHX 2% led to CFU-reductions by $\geq 6$ log₁₀ steps irrespective of the culture period, while CHX 0.2% reduced the CFU by $\geq 4$ log₁₀ steps in 48 h biofilms and by $\geq 3$ log₁₀ steps in 72 h biofilms. UVC radiation led to reductions by $\geq 6$ log₁₀ steps for radiation doses of 0.065 J/cm², 0.13 J/cm² or 0.26 J/cm².

## Spectroscopic measurements for release of nucleic acids upon CAP treatment

For assessing damage of cytoplasmic membranes upon CAP treatment, the release of nucleic acids from the cytoplasm was measured spectroscopically at 260 nm. Results are depicted in Fig 5. While the positive control showed a clear increase in median OD (0.5525) as compared to untreated controls (median OD 0.012), CAP treatment showed no increases in OD (median OD < 0.1), irrespective whether treatment was for 5 or 10 min. These results imply that the course of inactivation does not include direct cell lysis during CAP treatment.

## Discussion

Cold atmospheric plasma generates a wide variety of reactive species in form of *e.g.* ozone, reactive oxygen species and nitrogen dioxide which are the main source for its antimicrobial efficacy [40]. These CAP-derived reactive species are generated by a contact-free manner and are also not highly specialized to different bacterial genera or strains. This is a major advantage in comparison to antibiotics which target only specific structures according to the so-called key-lock principle [41]. That is the crucial aspect in which CAP could be named superior to common antibiotics and antiseptics, which are countered by bacteria through specific mechanisms of resistance [42]. Up to now, the risk of bacteria developing resistance against CAP is considered minor even after multiple sublethal treatments due to the point that no specific bacterial target is in focus but rather multiple targets [14].

In general, two approaches are in the focus with regard to CAP. The indirect method is using an agent to submit the effective particles as it is used in plasma jets or SMD technology [43–45]. Such a device consists of an outer electrode and an inner electrode and the gas is pumped through and thereby ionized. This method allows a longer distance between the affected area and the CAP device [46, 47]. Such devices were already used for treating skin diseases to reduce the microbial colonization of the infected tissue. SMD technology as it is used in this study is based on the same approach as the plasma jet, but a constant flow of externally applied gas is not needed because it uses ambient air as its carrier gas. The direct method submits the effective particle directly to the surface of the treated object (*e.g.* patient), that means the patient operates as the counter-electrode during treatment. Limitation is the distance between operating tool and the object of treatment [48]. Joaquin *et al.* used an atmospheric plasma jet to effectively inactivate a biofilm produced by *Chromobacterium violaceum* [49]. Laroussi *et al.* reviewed an atmospheric plasma jet and distinct discharge plasma sources to inactivate different bacterial species [50]. Depending on the chemical composition of reactive species generated by CAP, it was demonstrated that high levels of oxygen improved the antibacterial efficacy, *e.g.* towards *Escherichia coli [51]*. Furthermore, inactivation of extracellular virulence factors by CAP is also possible. Ziuzina *et al.* showed that inactivation of different virulence factors like pyocyanin or other quorum sensing molecules in *Pseudomonas aeruginosa* was successful [52]. In addition, the cytotoxic effects of pyocyanin were as well effectively reduced by CAP. So far, different studies were performed with plasma devices like plasma jets

**Table 1. Results from antimicrobial assay towards *E. faecalis* biofilms cultured for 48 h or 72 h.**

| Treatment | Concentration / radiation dose | Treatment period | Biofilm age | |
|---|---|---|---|---|
| | | | 48 h | 72 h |
| CAP | n/a | 1 min | 1.8 | 1.7 |
| | n/a | 3 min | 1.8 | 2.0 |
| | n/a | 5 min | 2.6 | 2.4 |
| | n/a | 10 min | 5.7 | 4.9 |
| CHX | 0.2% | 5 min | 4.1 | 3.9 |
| | 2% | 5 min | $\geq 6$ | $\geq 6$ |
| UVC radiation | 0.065 J/cm$^2$ | n/a | $\geq 6$ | - |
| | 0.13 J/cm$^2$ | n/a | $\geq 6$ | - |
| | 0.26 J/cm$^2$ | n/a | $\geq 6$ | - |

n/a = not applicable

- = not determined

This table shows the results from antimicrobial assay towards *E. faecalis* biofilms cultured for 48 h or 72 h. Values show $\log_{10}$ reductions

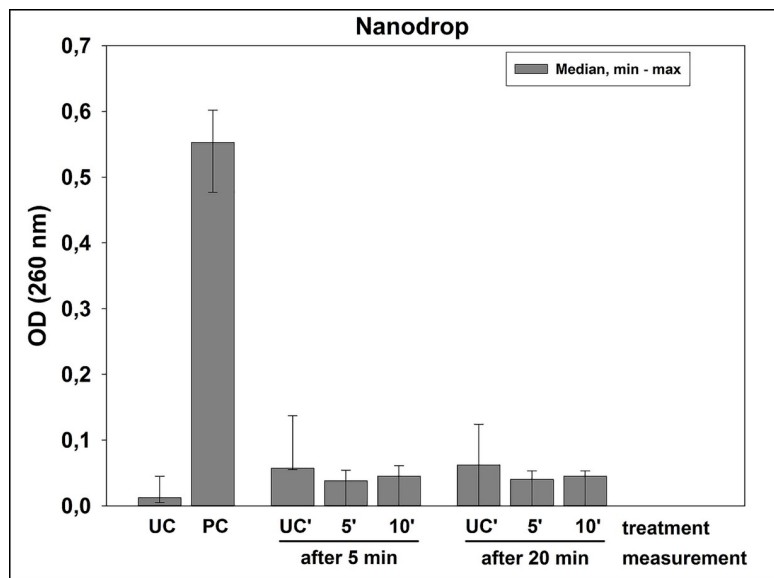

**Fig 5. Spectroscopic measurements for release of nucleic acids upon CAP treatment.** OD medians, minima and maxima of the supernatants of untreated biofilms (UC) or biofilms treated with positive control (lysozyme treatment followed by Proteinase K digestion, (PC)) or CAP, as measured at 260 nm for release of nucleic acids. For CAP, samples were measured either 5 or 20 min after being treated with CAP for 5 or 10 min or left untreated (UC'). $n = 6$.

and direct dielectric cold plasma devices, to affect *E. faecalis*, as organism of the ESKAPE pathogen group [53, 54]. Modic *et al.* showed the efficacy of CAP against biofilms formed by *Staphylococcus aureus* and *Pseudomonas aeruginosa*, which also belong to the ESKAPE pathogen group [55]. Moreover, other studies also demonstrated the antibacterial efficacy of cold plasmas against ESKAPE pathogens [56, 57]. Application of CAP against *E. faecalis* yielded promising effects, also in other environments (like in root canal treatment) when compared to photodynamic inactivation of microorganisms [58, 59]. Du *et al.* treated monospecies biofilms of *E. faecalis* with CAP [60]. They showed that the antibacterial efficacy was highest within the first 3 min and decreases after 10 and 30 min. In addition, they were able to confirm that the antibacterial efficacy of their plasma device was similar to that of 2% CHX [60]. Therefore, our study model is in accordance with already published papers, that *E. faecalis* grown as a biofilm can be successfully inactivated within a treatment period of up to 10 min. It is important to clarify that no effects on host cells and tissues occur with the use of the SMD device, since it has already been shown that CAP generated from this SMD technology exhibited no mutagenicity or toxicity against host cells and tissues *in vitro* or *ex vivo* [61, 62].

The first goal of this study was to prove that a new CAP device can kill *E. faecalis* seeded as planktonic cultures on agar plates. Here, a CFU-reduction of $> 7 \log_{10}$ steps was found after 1 min of CAP treatment, while longer treatment periods showed even higher reductions of CFU. Based on these results, the next experiments were focused on biofilms formed by *E. faecalis*. Bacteria in biofilms are generally more tolerant against antimicrobial approaches, which poses health risks in all fields of medicine, including dentistry [9]. Accordingly, *E. faecalis* biofilms are known to be problematic to handle in the field of endodontics and have been subject to different studies in the past [24, 63]. In the present study, biofilms of *E. faecalis* were cultured for different periods of time to investigate potential effects of the biofilm culture period on the antimicrobial efficacy of this CAP device. In 24 h biofilms, CAP treatment showed CFU-reductions by $\geq 3 \log_{10}$ after 5 min and $\geq 5 \log_{10}$ after 10 min. CFU-reductions by $\geq 5 \log_{10}$ steps could also be achieved in 48 h and 72 h old biofilms when they were subjected to

CAP treatment for 10 min. Herbst *et al.* infected 50 root canals with *E. faecalis* and used different methods for antimicrobial treatment CFU [63]. The most effective methods were CHX combined with CAP followed by CHX and CAP alone. The results showed that the efficacy against *E. faecalis* was in the same range for CHX and CAP. Li *et al.* cultured biofilms of *E. faecalis* in root canals over 3 weeks and subjected them to CAP or CHX [24]. The results again showed similar results between CHX and CAP. After a treatment (either with CAP or CHX) for 12 min, the number of CFU was under the limit of detection. As mentioned above, the two studies by Herbst *et al.* and Li *et al.* showed that CAP was as effective as CHX for killing of *E. faecalis* biofilms. Here, CHX was slightly more effective irrespective of biofilm culture period. However, it must be considered that CAP works without any physical contact and may therefore be superior to CHX in terms of handling and application. In this instance, it is also noteworthy that the CAP device used in this study works with ambient air as a carrier gas in contrast to other plasma devices, which mostly use Argon or Helium as a carrier gas [62].

In the third part of this study, we tried to get first insights into the antimicrobial mechanism of CAP. Up to this point the mechanism of different CAP treatments were investigated for *Escherichia coli* and *Staphylococcus aureus* with different results [64]. Han *et al.* showed that both bacterial strains reacted in different ways to the exposition of CAP. *E. coli* was inactivated by cell leakage due to the damage of the lipopolysaccharides in the outer cell membrane and the thin peptidoglycan layers of the cell wall of Gram-negative bacteria. *S. aureus* on the other hand contains very thick peptidoglycan layers as Gram-positive bacteria compared to *E. coli*. Here, damage of intracellular components was observed, but the cell wall was still intact [64]. Therefore, we aimed to determine whether damage of cytoplasmic membranes may be the primary antimicrobial mechanism of CAP generated by SMD technology. For this purpose, release of nucleic acids upon treatment was measured spectroscopically at 260 nm [37]. The results showed that there was no increase in OD upon treatment with CAP, even after a duration of 5 or 20 min between CAP treatment and measurement. Therefore, the antimicrobial effects of CAP may not be due to direct damage of cytoplasmic membranes and the subsequent release of cytoplasmic constituents. That indicates that the generated reactive species may rather lead to a malfunction than to a disruption of the cell membrane by Gram-positive bacterial species. Recently, our group could demonstrate that direct damage of DNA seems not to be the primary mechanism of action of CAP due to the point that *D. radiodurans* could be eradicated efficiently by CAP [29]. *D. radiodurans* contains a very efficient DNA repair mechanism to survive hundreds of DNA double strand breaks by reassembling these accurately before initiation of the next cycle of cell division [65]. The effective inactivation of this bacterium by CAP indicates that bacterial targets other than DNA are damaged by CAP, since *D. radiodurans* can repair DNA damage more efficiently than *E. faecalis* or other bacteria [66]. Recently, Arjunan *et al.* summarized the current knowledge about the interactions of reactive oxygen species and reactive nitrogen species with DNA and its components generated by CAP devices, and reported that strand breaks in DNA or peroxynitrite oxidation of nucleotides take place [67]. Overall, we could show that this new CAP device has a pronounced antimicrobial efficacy against *E. faecalis* on agar plates as well as in biofilms and we could give insight into the mechanisms of action of CAP in terms of not releasing DNA and therefore not the possible problems induced by DNA damage. However, further studies on the antimicrobial mechanisms of CAP are essential for deeper understanding the effects of CAP.

## Conclusions

In this study cold atmospheric plasma was used to achieve an antibacterial reduction of >99.9% of *Enterococcus faecalis* growing as biofilms similarly to the effect of 0.2%

chlorhexidine. The superiority of CAP is the fact that it operates completely contact free. Furthermore, this study demonstrated that release of cytoplasmic components due to direct damage of the bacterial cell wall or membrane seems not to be the primary mechanism of inactivation of *E. faecalis* by CAP.

## Supporting information

**S1 File. Data sets necessary to replicate study findings for Fig 2, Fig 3, Fig 4, Fig 5 and Table 1 are listed.**
(DOCX)

## Acknowledgments

This study was funded by the grant "BayMed: 41-6618c/272/1-MED-1507-0004" (Bayern Innovativ GmbH, Germany). Furthermore, Fabian Cieplik thanks for funding by the University Medical Center Regensburg (ReForM B program) and the Deutsche Forschungsgemeinschaft (DFG, German Research Foundation; grant CI 263/1-3). Denise Muehler is gratefully acknowledged for her valuable help with the DNA release experiments.

## Author Contributions

**Conceptualization:** Fabian Cieplik, Tim Maisch, Julia L. Zimmermann.

**Data curation:** Felix Theinkom.

**Formal analysis:** Karl-Anton Hiller.

**Funding acquisition:** Sylvia Cantzler, Julia L. Zimmermann.

**Investigation:** Felix Theinkom, Larissa Singer, Fabian Cieplik, Tim Maisch.

**Methodology:** Felix Theinkom, Larissa Singer, Sylvia Cantzler, Hannes Weilemann, Maximilian Cantzler, Karl-Anton Hiller.

**Project administration:** Sylvia Cantzler, Hannes Weilemann, Maximilian Cantzler, Tim Maisch, Julia L. Zimmermann.

**Resources:** Hannes Weilemann, Maximilian Cantzler.

**Software:** Hannes Weilemann, Maximilian Cantzler, Karl-Anton Hiller.

**Supervision:** Fabian Cieplik, Karl-Anton Hiller, Tim Maisch, Julia L. Zimmermann.

**Validation:** Karl-Anton Hiller, Tim Maisch, Julia L. Zimmermann.

**Writing – original draft:** Felix Theinkom, Larissa Singer, Tim Maisch, Julia L. Zimmermann.

**Writing – review & editing:** Fabian Cieplik, Sylvia Cantzler, Karl-Anton Hiller, Tim Maisch, Julia L. Zimmermann.

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
