## [Decision Letter · Decision Letter 0]

22 Jul 2019

PONE-D-19-18041

Antibacterial efficacy of cold atmospheric plasma against Enterococcus faecalis planktonic cultures and biofilms in vitro

PLOS ONE

Dear Dr. Maisch,

Thank you for submitting your manuscript to PLOS ONE. After careful consideration, we feel that it has merit but does not fully meet PLOS ONE’s publication criteria as it currently stands. Therefore, we invite you to submit a revised version of the manuscript that addresses the points raised during the review process.

We would appreciate receiving your revised manuscript by Sep 05 2019 11:59PM. To enhance the reproducibility of your results, we recommend that if applicable you deposit your laboratory protocols in protocols.io, where a protocol can be assigned its own identifier (DOI) such that it can be cited independently in the future. For instructions see: http://journals.plos.org/plosone/s/submission-guidelines#loc-laboratory-protocols

We look forward to receiving your revised manuscript.

Kind regards,

Monica Cartelle Gestal, PhD

Academic Editor

PLOS ONE

3. Thank you for stating the following in the Financial Disclosure section: "The authors Sylvia Cantzler, Hannes Weilemann, Maximilian Cantzler are employees from terraplasma GmbH (Garching, Germany), and Julia L. Zimmermann is an employee from terraplasma medical GmbH (Garching, Germany), which have developed the plasma device used in this study. All other authors declare no conflicts of interests. The funders had no role in study design, data collection and analysis, decision to publish, or preparation of the manuscript.

This study was funded by the grant “BayMed: 41-6618c/272/1-MED-1507-0004” (Bayern Innovativ GmbH, Germany). Furthermore, Fabian Cieplik thanks for funding by the University Medical Center Regensburg (ReForM B program). Denise Muehler is gratefully acknowledged for her valuable help with the DNA release experiments."

We note that one or more of the authors have an affiliation to the commercial funders of this research study: terraplasma GmbH and terraplasma medical GmbH.

a). Please provide an amended Funding Statement declaring this commercial affiliation, as well as a statement regarding the Role of Funders in your study. If the funding organization did not play a role in the study design, data collection and analysis, decision to publish, or preparation of the manuscript and only provided financial support in the form of authors' salaries and/or research materials, please review your statements relating to the author contributions, and ensure you have specifically and accurately indicated the role(s) that these authors had in your study. You can update author roles in the Author Contributions section of the online submission form.

b). Please also provide an updated Competing Interests Statement declaring this commercial affiliation along with any other relevant declarations relating to employment, consultancy, patents, products in development, or marketed products, etc.  

Additional Editor Comments (if provided):

This manuscript is of interest, however, there are some comments that our reviewers highlighted and that need to be address prior to acceptance of the paper.

Reviewers' comments:

Reviewer's Responses to Questions

**Comments to the Author**

1. Is the manuscript technically sound, and do the data support the conclusions?

Reviewer #1: Yes

Reviewer #2: Partly

2. Has the statistical analysis been performed appropriately and rigorously? 

Reviewer #1: Yes

Reviewer #2: No

3. Have the authors made all data underlying the findings in their manuscript fully available?

Reviewer #1: Yes

Reviewer #2: Yes

4. Is the manuscript presented in an intelligible fashion and written in standard English?

Reviewer #1: Yes

Reviewer #2: No

5. Review Comments to the Author

Reviewer #1: Although the idea of using cold plasmas for antimicrobial therapy is not new, the paper is dealing with an important and well investigated subject, antimicrobial effect of such plasmas against one of the ESKAPE pathogens, E.faecalis. This approach could be eficiently used for the therapy of wounds and also in dentistry.

The results are clearly presented and conclusions are supported by experimental data.

Iwould reccomend authors to expand their discussion by adding some details regarding the known effect of cold plasmas against other ESKAPE pathogens, highlighting how would this approach improve the therapy of such infections. Also discussing the antimicrobial mechanisms of cold plasmas against other microbial species, could help in revealing the primary antimicrobial mechanism against E.faecalis.

Reviewer #2: The authors focus to demonstrate possible antimicrobial activity of CAPs on E. faecalis. The authors used CAP in both planktonic and biofilms to investigate the possible antimicrobial effects. The authors reported a decrease in CFUs after 5 and 10 minutes of CAP treatment in 24h and 72h biofilms. The authors suggest that the antimicrobial activity of CAP is not due to damage to cell membranes as they did not observe any DNA release after treatment.

My main concerns of the presented manuscript are listed below:

1. The authors failed to explain and elaborate on the importance and possible future application of antimicrobial activity of CAP on E. faecalis. The addition of more elaborate details on the rational of choosing this bacteria to test the antimicrobial activity of CAPs would improve the quality of the introduction.

2. Although the details of experimental plans are explained thoroughly, there are no explanation of observed results in the result section.

3. The statistical analysis are not explained and the tests are not mentioned and the levels of significance are not reported.

4. the details of the CAP used in this study is included in discussion and if moved to introduction, clarifies the methods and experimental plans in the study. The importance and applications suggested in the discussion can not be withdrawn from the results presented in this study are ambiguous and require further studies that shows no negative effects on host cells and tissues.

5. Although the experiments are performed in an environment similar to human saliva, there are no explanations of the importance of use of this environment and whether other environments may impact on level of antimicrobial efficacy of CAPs.

6. PLOS authors have the option to publish the peer review history of their article (what does this mean?). If published, this will include your full peer review and any attached files.

Reviewer #1: Yes: Alina Maria Holban

Reviewer #2: No

---

## [Author Response · Author response to Decision Letter 0]

16 Aug 2019

Rebuttal letter to the reviewers (a separate file was uploaded)

Review Comments to the Author

Reviewer #1: Although the idea of using cold plasmas for antimicrobial therapy is not new, the paper is dealing with an important and well investigated subject, antimicrobial effect of such plasmas against one of the ESKAPE pathogens, E. faecalis. This approach could be efficiently used for the therapy of wounds and also in dentistry.

The results are clearly presented and conclusions are supported by experimental data.

I would recommend authors to expand their discussion by adding some details regarding the known effect of cold plasmas against other ESKAPE pathogens, highlighting how would this approach improve the therapy of such infections. Also discussing the antimicrobial mechanisms of cold plasmas against other microbial species, could help in revealing the primary antimicrobial mechanism against E.faecalis.

All changes are marked in yellow within the manuscript (file labeled as “Revised Manuscript with Track Changes”).

Answer:

We thank the reviewer for the comments. We have inserted some new statements concerning efficacy of CAP against ESKAPE pathogens within the discussion section (page 17, line 365-378). Here in this study we have decided to use E. faecalis due to the fact that this pathogen is known to grew and form biofilms on both inanimate and living surfaces and therefore E. faecalis gained tolerance and susceptibility against external influences from the environment (page 19, line 426-432).

 

Reviewer #2: The authors focus to demonstrate possible antimicrobial activity of CAPs on E. faecalis. The authors used CAP in both planktonic and biofilms to investigate the possible antimicrobial effects. The authors reported a decrease in CFUs after 5 and 10 minutes of CAP treatment in 24h and 72h biofilms. The authors suggest that the antimicrobial activity of CAP is not due to damage to cell membranes as they did not observe any DNA release after treatment.

All changes are marked in green within the manuscript. (file labeled as “Revised Manuscript with Track Changes”).

My main concerns of the presented manuscript are listed below:

1. The authors failed to explain and elaborate on the importance and possible future application of antimicrobial activity of CAP on E. faecalis. The addition of more elaborate details on the rational of choosing this bacteria to test the antimicrobial activity of CAPs would improve the quality of the introduction.

Answer:

Here in this study we used E. faecalis as one the ESKAPE pathogens which is known to be able to grew and form biofilms on both inanimate and living surfaces and therefore E. faecalis gained tolerance and susceptibility against external influences from the environment. We have added new statements on page 3, line 61-64.

2. Although the details of experimental plans are explained thoroughly, there are no explanation of observed results in the result section.

Answer:

We have now revised the results section as recommended.

3. The statistical analysis are not explained and the tests are not mentioned and the levels of significance are not reported.

Answer:

We thank the reviewer for the comment. Indeed, we did not explain the statistical test and did not mention the level of significance, because we did not perform classic significance analyses. However, we mentioned in the data analysis section our definition of antimicrobial efficacy and disinfection. (L230-234: This passage reads as follows: “Medians on or below these lines mean a bacterial reduction by 3 log10 (≥99.9%) or by 5 log10 (≥99.999%). According to the guideline of infection control this means a biologically relevant antimicrobial activity or a disinfectant effect, respectively [Boyce JM, Pittet D. Guideline for Hand Hygiene in Health-Care Settings: recommendations of the Healthcare Infection Control Practices Advisory Committee and the HICPAC/SHEA/APIC/IDSA Hand Hygiene Task Force. Infect Control Hosp Epidemiol. 2002;23(12 Suppl):S3-40.]”).

From microbiological point of view concerning these definitions of the wordings “antimicrobial” and “disinfection”, we used these definitions as the biological relevant tool to discriminate the data.

In addition, thoroughly reviewing our data analysis data section again, we found that the spectroscopic measurements are depicted as median min and max (see fig. 5) and not as medians, first and third quartile. We changed this part accordingly within the data analysis section.

4. the details of the CAP used in this study is included in discussion and if moved to introduction, clarifies the methods and experimental plans in the study. The importance and applications suggested in the discussion cannot be withdrawn from the results presented in this study are ambiguous and require further studies that shows no negative effects on host cells and tissues.

Answer:

The SMD technology and the respective studies on the topic of mutagenicity against host cells and tissues are now included in the discussion section (page 18, line 389-393).

5. Although the experiments are performed in an environment similar to human saliva, there are no explanations of the importance of use of this environment and whether other environments may impact on level of antimicrobial efficacy of CAPs.

Answer:

The human saliva was used based on the studies performed by Pratten et al. which showed that usage of saliva is requisite for growth of a robust biofilm and this method was also used in different studies before as follows: 

i) Pratten J et al.: In vitro studies of the effect of antiseptic-containing mouthwashes on the formation and viability of Streptococcus sanguis biofilms. J Appl Microbiol. 1998;84(6):1149-55; 

ii) ii) Cieplik F et al.: The impact of absorbed photons on antimicrobial photodynamic efficacy. Front Microbiol. 2015;6:706; 

iii) iii) Cieplik F et al.: Photodynamic biofilm inactivation by SAPYR--an exclusive singlet oxygen photosensitizer. Free Radic Biol Med. 2013;65:477-87.

---

## [Decision Letter · Decision Letter 1]

2 Oct 2019

Antibacterial efficacy of cold atmospheric plasma against Enterococcus faecalis planktonic cultures and biofilms in vitro

PONE-D-19-18041R1

Dear Dr. Maisch,

We are pleased to inform you that your manuscript has been judged scientifically suitable for publication and will be formally accepted for publication once it complies with all outstanding technical requirements.

With kind regards,

Monica Cartelle Gestal, PhD

Academic Editor

PLOS ONE

Additional Editor Comments (optional):

Dear Dr. Maisch,

I am please to announce that the manuscript has been accepted in its current form.

Best regards,

Reviewers' comments:

Reviewer's Responses to Questions

**Comments to the Author**

1. If the authors have adequately addressed your comments raised in a previous round of review and you feel that this manuscript is now acceptable for publication, you may indicate that here to bypass the “Comments to the Author” section, enter your conflict of interest statement in the “Confidential to Editor” section, and submit your "Accept" recommendation.

Reviewer #2: All comments have been addressed

2. Is the manuscript technically sound, and do the data support the conclusions?

Reviewer #2: Yes

3. Has the statistical analysis been performed appropriately and rigorously? 

Reviewer #2: Yes

4. Have the authors made all data underlying the findings in their manuscript fully available?

Reviewer #2: Yes

5. Is the manuscript presented in an intelligible fashion and written in standard English?

Reviewer #2: Yes

6. Review Comments to the Author

Reviewer #2: (No Response)

7. PLOS authors have the option to publish the peer review history of their article (what does this mean?). If published, this will include your full peer review and any attached files.

Reviewer #2: No

---

## [Editor Report · Acceptance letter]

18 Nov 2019

PONE-D-19-18041R1 

Antibacterial efficacy of cold atmospheric plasma against Enterococcus faecalis planktonic cultures and biofilms *in vitro*

Dear Dr. Maisch:

I am pleased to inform you that your manuscript has been deemed suitable for publication in PLOS ONE. Congratulations! Your manuscript is now with our production department. 

With kind regards,

on behalf of

Dr. Monica Cartelle Gestal 

Academic Editor

PLOS ONE